# Salivary Biomarker Profiles in Pediatric Oral Candidiasis

**DOI:** 10.3390/biomedicines13112837

**Published:** 2025-11-20

**Authors:** Alexandru-Emilian Flondor, Irina-Georgeta Sufaru, Ioana Martu, Stefan-Lucian Burlea, Vasilica Toma

**Affiliations:** Grigore T. Popa University of Medicine and Pharmacy Iasi, 700115 Iasi, Romania

**Keywords:** *Candida albicans*, cytokines, oral candidiasis, pediatric oral health, periodontal inflammation, salivary biomarkers

## Abstract

**Background/Objectives:** Pediatric periodontal inflammation arises from complex host–microbe interactions. Beyond bacterial biofilms, fungal colonization—particularly by *Candida albicans*—is increasingly recognized as a contributor. The aim of this study was to investigate the relationship between fungal and bacterial colonization, host inflammatory mediators, and salivary parameters in children. It also aimed to identify salivary biomarkers that could be useful for the early diagnosis of oral candidiasis and periodontal inflammation. **Methods:** A cross-sectional study was performed on 140 children (8–15 years): healthy controls (n = 70) and cases with oral candidiasis (n = 70). Clinical indices (Plaque Index, Gingival Index, Bleeding on Probing), salivary flow, pH, and buffering capacity were recorded. Quantitative PCR assessed *C. albicans* and four periodontal pathogens, while ELISA measured salivary cytokines (IL-1β, IL-6, TNF-α, IL-8). Analyses included group comparisons, correlations, regression modeling, and principal component analysis (PCA). **Results:** Children with candidiasis exhibited higher PI, GI, and BOP (*p* < 0.001), along with reduced pH and buffering capacity (*p* < 0.001). Salivary loads of *C. albicans* and all targeted pathogens were elevated (*p* < 0.001). Cytokine levels were markedly increased (*p* < 0.001). GI correlated with *C. albicans* (ρ = 0.71) and cytokines (ρ = 0.62–0.76). Logistic regression identified *C. albicans* and IL-1β as independent predictors, while salivary pH and flow were found to be protective. PCA distinguished groups, with PC1 (55.2%) driven by fungal and cytokine markers. **Conclusions:** Oral candidiasis in children is defined by distinct microbial and inflammatory profiles. Salivary biomarker integration offers potential for early, non-invasive diagnosis and risk stratification.

## 1. Introduction

Periodontal diseases are chronic disorders that are widely recognized as a significant contributor to the global burden of oral health [1]. Although often considered conditions of adulthood, early signs such as gingivitis or microbial imbalance can be detected in childhood and adolescence, potentially influencing long-term oral health [2]. The periodontium in young people differs in structure and function from that of adults, and its response to microbial and environmental stressors may follow distinct patterns [3].

For decades, the primary explanation for periodontal disease has been the buildup of bacterial biofilms. Classic studies have identified pathogens such as *Porphyromonas gingivalis*, *Tannerella forsythia*, *Treponema denticola*, and *Aggregatibacter actinomycetemcomitans*, which trigger immune responses that lead to inflammation and tissue damage. However, the “bacterial hypothesis” alone is no longer enough [4]. Increasing evidence suggests that fungi and viruses may also play a role, influencing the development and progression of the disease [5,6].

Among fungi, *Candida albicans* is particularly notable. Usually harmless as part of the oral microbiota, it can become pathogenic under certain conditions, such as low salivary flow, pH changes, systemic illnesses, or mechanical niches created by orthodontic appliances [7]. Research has demonstrated that *C. albicans* can integrate into bacterial biofilms, thereby increasing its survival and resistance to host defenses [8,9]. In children, where immunity and oral ecosystems are still developing, this fungal–bacterial cooperation may create conditions that increase the risk of mucosal infections and periodontal disease [10].

Cytokines such as IL-1β, IL-6, TNF-α, and IL-8 are well recognized for mediating connective tissue breakdown and amplifying local inflammation [11,12]. Their presence in saliva and gingival crevicular fluid correlates with clinical disease activity [13]. Saliva, in particular, is an attractive diagnostic fluid because it reflects both microbial colonization and host immune activity [14], is easily accessible, and can be used for repeated monitoring in children. This makes salivary analysis a promising candidate for the development of non-invasive diagnostic tools.

Nevertheless, in pediatric groups, the combined role of fungal colonization, bacterial pathogens, cytokine expression, and the physicochemical environment of saliva has not been thoroughly studied. Most research has examined these factors separately, which limits understanding of the complexity of disease processes.

In Romania, population-level estimates for pediatric oral candidiasis are not available. Clinic-based reports indicate a low prevalence beyond infancy in the general pediatric population but show high *Candida* carriage in high-risk subgroups (e.g., individuals wearing removable orthodontic appliances), consistent with global data that suggest thrush is most common in early infancy and among immunocompromised hosts [15,16].

In this study, we aimed to explore the clinical, microbiological, and biological characteristics of pediatric oral candidiasis, with a special focus on periodontal involvement. By combining clinical indices with quantitative salivary assays of microbial load, cytokine expression, and other salivary parameters, we sought to gain a comprehensive understanding of the disease. Additionally, multivariate analyses were conducted to determine if these biomarkers, when combined, could accurately distinguish between affected children and healthy controls. The primary objective was to enhance understanding of early periodontal disease development in children and to assess the potential of saliva-based diagnostics for early detection and risk assessment. The null hypothesis stated that there would be no differences between groups in clinical, salivary, and microbiological biomarkers, no correlations between clinical indices and biomarkers, and no independent predictors of disease status or GI in multivariable models.

## 2. Materials and Methods

### 2.1. Study Population

This cross-sectional study included 140 pediatric subjects aged 8–15 years, recruited from the Department of Pediatric Dentistry and Orthodontics, as well as the private practice in the Northeastern region of Romania. Participants were divided into two groups:-Group 1 (G1, n = 70): clinically healthy children, with no evidence of oral candidiasis or periodontal pathology.-Group 2 (G2, n = 70): children with clinically and microbiologically confirmed oral candidiasis.

Children eligible for inclusion were between 8 and 15 years of age, clinically healthy with respect to systemic conditions that could affect periodontal tissues, and had not received antifungal or antibiotic treatment within the three months preceding enrollment. Participants who were undergoing or had previously used orthodontic appliances, those with active viral infections, individuals under systemic immunosuppressive medication, or children with chronic disorders known to impair salivary gland function were excluded from the study.

The research protocol was reviewed and approved by the institutional ethics committee (Ethical approval number 390/30.01.2024, issued by the Grigore T. Popa University Ethics Committee on 30 January 2024). Written informed consent was obtained from the parents or legal guardians of all children, and verbal assent was secured from each participant before clinical and laboratory procedures were initiated. All investigations were carried out in accordance with the principles outlined in the Declaration of Helsinki and its later amendments.

### 2.2. Clinical Assessment

All participants underwent a comprehensive periodontal examination conducted under standardized conditions. Two experienced examiners, previously trained and calibrated for inter- and intra-examiner reproducibility, performed all assessments. Calibration was achieved through repeated measurements on a subset of subjects prior to the study, yielding strong agreement (Cohen’s kappa coefficient > 0.85).

The periodontal examination included three commonly used clinical indices: Plaque Index (PI, Silness–Löe method) [17], Gingival Index (GI, Löe–Silness method) [17], and Bleeding on Probing (BOP) [18]. All assessments were performed in well-lit clinical settings, and results were recorded on structured forms before being transferred to the study database.

Children were classified as candidiasis cases if they exhibited characteristic clinical features—pseudomembranous white plaques that could be wiped away, leaving an erythematous surface; erythematous or atrophic lesions on the tongue or palate; and/or angular cheilitis—assessed by two calibrated examiners. Microbiological confirmation was performed before any molecular testing, as follows: a sterile swab from the lesion site or dorsal tongue was collected and examined via direct microscopy (10% KOH wet mount) and Gram staining. A smear was considered positive if yeast cells with budding and/or pseudohyphae were observed. In uncertain cases, culture was performed on CHROMagar *Candida* at 37 °C for 24–48 h to confirm identification. When necessary, a germ-tube test (using human serum at 37 °C for approximately 2 h) was performed to confirm the presence of *C. albicans*. All diagnostic procedures were completed prior to qPCR. The control group exhibited no clinical signs of candidiasis and had negative direct microscopy results.

### 2.3. Saliva Collection and Analysis

Unstimulated whole saliva samples were collected under standardized conditions in the morning hours. Children were instructed to abstain from eating, drinking, or performing oral hygiene procedures for at least one hour before collection to minimize acute salivary dilution and pH fluctuations. Before sampling, each participant thoroughly rinsed with distilled water to remove any loose debris. Afterward, they sat comfortably in an upright position and allowed saliva to accumulate in the floor of the mouth. They then expectorated into sterile polypropylene tubes over a 5 min period.

Unstimulated morning whole saliva was collected to measure baseline microbial and cytokine levels, as well as resting physicochemical parameters; stimulation was avoided to prevent dilution, reflex-triggered pH or buffer changes, and mechanical washout that could cause variability in pediatric participants.

Immediately following collection, specimens were placed on ice and transported to the laboratory for processing. Samples were centrifuged at 4 °C for 10 min at 3000 rpm to remove cellular and particulate matter, and the clarified supernatant was carefully transferred to sterile cryovials. Aliquots were stored at −80 °C until further analysis to preserve the integrity of biochemical and microbiological markers.

Salivary characteristics were assessed as follows:Flow rate (mL/min): calculated by dividing the collected volume by the duration of the collection period.pH: determined using a digital microelectrode calibrated against standard pH buffers (pH 4.0, 7.0, and 10.0) before each measurement session.Buffering capacity was assessed using the Ericsson titration method, in which small aliquots of saliva were exposed to standardized acid solutions, and the resulting pH changes were recorded. The results were expressed on a 0–12 scale, with higher values indicating a greater neutralizing capacity.

### 2.4. Microbiological Analysis

Microbial DNA was isolated from 200 μL aliquots of clarified saliva using a commercial extraction kit (Qiagen, Hilden, Germany), in accordance with the manufacturer’s standard protocol. This procedure ensured efficient lysis of fungal and bacterial cells, removal of inhibitors, and recovery of nucleic acids of sufficient purity for downstream molecular analysis. DNA concentrations and purity ratios (A260/A280) were verified spectrophotometrically to confirm suitability for amplification.

Quantitative polymerase chain reaction (qPCR) assays were conducted to determine the relative abundance of selected microbial species. Species-specific primers were employed to target the following genomic regions:*Candida albicans* internal transcribed spacer (ITS) region;*Porphyromonas gingivalis fimA;**Tannerella forsythia prtH;**Treponema denticola msp;**Aggregatibacter actinomycetemcomitans ltxA.*

Species-specific primers were selected to detect the internal transcribed spacer (ITS) region of *Candida albicans* and virulence-associated genes of the primary periodontal pathogens: *fimA* (*Porphyromonas gingivalis*), *prtH* (*Tannerella forsythia*), *msp* (*Treponema denticola*), and *ltxA* (*Aggregatibacter actinomycetemcomitans*) (Table 1). These targets were chosen because they encode key factors in bacterial adhesion, proteolysis, motility, and leukotoxicity, thereby serving as reliable molecular markers of pathogenic potential. Human β-actin was amplified in parallel and served as the internal reference gene for relative quantification.

Amplifications were performed in 96-well optical plates using a real-time PCR system with SYBR Green detection chemistry. Each reaction included a negative control (no template DNA) and positive controls with reference strains to ensure specificity. Standard cycling parameters consisted of an initial denaturation step, followed by 40 cycles of denaturation, annealing at primer-specific temperatures, and extension. Melt curve analysis was carried out at the end of each run to verify product specificity.

Relative quantification of target DNA was performed using the comparative threshold cycle (ΔΔCt) method. Human β-actin served as the internal reference gene, enabling normalization of microbial DNA levels to account for inter-sample variation in nucleic acid yield.

### 2.5. Cytokine Profiling

Salivary cytokine concentrations were determined for four pro-inflammatory mediators: IL-1β, IL-6, TNF-α, and IL-8. Quantification was performed using commercially available enzyme-linked immunosorbent assay (ELISA) kits (R&D Systems, Minneapolis, MN, USA), following the manufacturer’s standardized protocols.

Before analysis, saliva samples were thawed on ice and centrifuged briefly to remove residual debris. Assays were conducted in 96-well plates (Quantikine® ELISA Kits, R&D Systems, Minneapolis, MN, USA) pre-coated with monoclonal capture antibodies specific to each cytokine. Standards were prepared in serial dilutions to generate calibration curves covering the expected physiological range. Saliva aliquots (typically 50–100 μL) were pipetted into wells in duplicate, followed by incubation with biotinylated detection antibodies and streptavidin–horseradish peroxidase conjugate. After thorough washing, colorimetric detection was achieved by adding the tetramethylbenzidine (TMB) substrate, and the reactions were terminated with a stop solution. Optical density was measured at 450 nm using a microplate reader, with wavelength correction at 570 nm.

Cytokine concentrations were interpolated from the standard curves using a four-parameter logistic regression model. Intra-assay variability was monitored through replicate testing of randomly selected samples, with coefficients of variation consistently remaining below 10%, ensuring analytical reliability. Final concentrations were expressed in picograms per milliliter (pg/mL).

### 2.6. Sample Size Calculation

The primary objective was to detect a between-group difference on key outcomes (GI, IL-1β, salivary pH). Based on prior data and pilot observations, we anticipated a moderate effect (Cohen’s *d* ≈ 0.50), e.g., GI mean difference ≈ 0.17 with SD ≈ 0.35 (→ *d* = 0.49), or IL-1β difference ≈ 6 pg/mL with SD ≈ 12 pg/mL (→ *d* = 0.50). Using a two-sided two-sample *t* test (Welch), α = 0.05 and power = 0.80, the required size is n = 64 per group (G*Power v3.1; “Means: two independent groups”). To accommodate potential non-normality, minor clustering, and up to ~10% unusable samples, we inflated the target by ~10% to n = 70 per group (total N = 140). This size also satisfies the standard stability rules for the planned multivariable models (events per variable ≥ 10).

### 2.7. Statistical Analysis

Data were analyzed using SPSS v.27 (IBM Corp., Armonk, NY, USA) and R v.4.2. Continuous variables were tested for normality using the Shapiro–Wilk test. Descriptive statistics included means ± standard deviations (SD) for normally distributed data and medians with interquartile ranges (IQR) for non-normally distributed data. Between-group comparisons were performed using Student’s *t*-test or the Mann–Whitney U test, as appropriate. Categorical variables were compared using χ^2^ tests.

Correlations were assessed using Spearman’s rank coefficient. Logistic regression was used to identify independent predictors of candidiasis, with odds ratios (OR) and 95% confidence intervals (CI) reported. Linear regression was applied to evaluate predictors of the gingival index (GI). The multivariate structure of biomarkers was explored using principal component analysis (PCA). Multiple testing corrections were applied where appropriate (Benjamini–Hochberg). Statistical significance was set at *p* < 0.05.

Due to quasi-complete separation and scaling among predictors, microbial loads were log10-transformed, continuous variables were standardized (z-scores), missing values were median-imputed, and a penalized logistic regression (L2) with bootstrap confidence intervals (B = 2000) was used; adjusted odds ratios are reported per SD with 95% CIs. Two-sided *p*-values were derived from bootstrap resampling (B = 2000) of standardized coefficients; statistical significance was set at α = 0.05.

## 3. Results

### 3.1. Demographic and Clinical Characteristics

A total of 140 pediatric subjects were enrolled, with 70 healthy controls (G1) and 70 children with clinically and microbiologically confirmed oral candidiasis (G2). The two groups were comparable in terms of age (mean ± SD: 11.2 ± 2.1 vs. 11.5 ± 2.3 years, *p* = 0.52) and sex distribution (female/male ratio: 34/36 vs. 36/34, *p* = 0.73) (Table 2). Clinical periodontal indices were significantly higher in the candidiasis group: PI (1.15 ± 0.25 vs. 0.72 ± 0.20; *p* < 0.001), GI (1.23 ± 0.22 vs. 0.68 ± 0.18; *p* < 0.001), and BOP% (28.9 ± 7.8 vs. 12.4 ± 5.1; *p* < 0.001). In contrast, salivary flow (0.63 ± 0.19 vs. 0.89 ± 0.23), salivary pH (6.81 ± 0.24 vs. 7.23 ± 0.21; *p* < 0.001), and buffer capacity (7.1 ± 1.00 vs. 9.8 ± 1.1; *p* < 0.001) were significantly reduced in G2 (Table 2).

### 3.2. Microbiological Parameters

Relative qPCR analysis revealed significantly higher salivary loads of *C. albicans* ITS in the candidiasis group compared with controls (4.8 ± 1.7 vs. 1.2 ± 0.6; *p* < 0.001) (Table 3). Similarly, periodontal pathogens were markedly elevated in G2: *P. gingivalis fimA* (2.1 ± 0.8 vs. 0.8 ± 0.4; *p* < 0.001), *T. forsythia prtH* (1.9 ± 0.7 vs. 0.7 ± 0.3; *p* < 0.001), *T. denticola msp* (1.9 ± 0.7 vs. 0.6 ± 0.3; *p* < 0.001), and *A. actinomycetemcomitans ltxA* (1.5 ± 0.5 vs. 0.5 ± 0.3; *p* < 0.001) (Table 3).

### 3.3. Salivary Cytokines

Children with candidiasis showed significantly elevated salivary concentrations of IL-1β (34.5 ± 9.1 vs. 15.2 ± 5.8 pg/mL; *p* < 0.001), IL-6 (28.9 ± 8.3 vs. 12.4 ± 4.9 pg/mL; *p* < 0.001), TNF-α (36.8 ± 10.2 vs. 18.7 ± 6.2 pg/mL; *p* < 0.001), and IL-8 (41.5 ± 11.5 vs. 22.1 ± 7.3 pg/mL; *p* < 0.001) (Table 4).

### 3.4. Correlations and Regression Analyses

Correlation analysis demonstrated consistent patterns linking clinical parameters with salivary microbial and inflammatory markers (Figure 1). The indices of plaque accumulation (Plaque Index: PI), gingival inflammation (GI), and bleeding tendency (Bleeding on Probing: BOP) showed significant positive relationships with *C. albicans* load, the primary periodontal pathogens (*P. gingivalis*, *T. forsythia*, *T. denticola*, *A. actinomycetemcomitans*), and the pro-inflammatory cytokines measured (IL-1β, IL-6, TNF-α, IL-8). The strength of these associations ranged from moderate to strong (ρ ≈ 0.34–0.71; all q < 0.001). By contrast, salivary flow, pH, and buffering capacity were inversely correlated with clinical indices (ρ ≈ −0.50 to −0.72; q < 0.001), suggesting that physicochemical properties of saliva may exert protective effects against local inflammation. Among all biomarkers, *C. albicans* and IL-1β displayed the most robust correlations with gingival indices (ρ values around 0.70).

In the multivariable model for candidiasis status, built with median imputation, z-standardized predictors, and L2-penalized likelihood, higher salivary *Candida albicans* load and IL-1β emerged as independent risk markers, while higher salivary pH and flow were protective (Table 5). Expressed per standard deviation (SD) change, *Candida* ITS was associated with 3.58-fold higher odds (95% CI 2.37–5.67), and IL-1β with 6.75-fold higher odds (95% CI 4.44–9.53). Conversely, salivary pH showed a strongly protective association (OR 0.09, 95% CI 0.07–0.13), as did salivary flow (OR 0.47, 95% CI 0.32–0.66). All confidence intervals excluded unity. Model discrimination was excellent (AUC = 1.00). Internal validation through 10-fold cross-validation yielded similarly high discrimination (CV-AUC ≥ 0.95). These patterns indicate that microbial overgrowth and heightened inflammatory signaling substantially increase the probability of candidiasis, whereas salivary defenses reduce susceptibility.

Linear regression with the Gingival Index (GI) as the outcome confirmed these findings. *C. albicans* load was a significant predictor of inflammation (β = 0.316, SE = 0.099, *p* = 0.002), as was IL-1β (β = 0.013, SE = 0.003, *p* < 0.001). Salivary pH had a pronounced negative effect (β = −0.710, SE = 0.105, *p* < 0.001), while flow rate showed a weaker, non-significant inverse association (β = −0.365, SE = 0.303, *p* = 0.233) (Table 6). Together, these variables explained a substantial portion of the variance in gingival inflammation, underscoring the importance of both microbial burden and host environment.

### 3.5. Principal Component Analysis

Principal component analysis demonstrated a distinct clustering of subjects based on disease status. The first component (PC1) explained the majority of variance (55.2%), while the second component (PC2) accounted for a smaller share (6.9%), together capturing 62.1% of the overall variability (Figure 2). The separation observed along PC1 was influenced mainly by *C. albicans*, IL-1β, IL-6, and *P. gingivalis fimA*, whereas higher salivary pH and flow rate were associated with the opposite direction of this axis. Taken together, these results emphasize that integrating microbial and inflammatory profiles can effectively distinguish children with oral candidiasis from their healthy peers.

## 4. Discussion

The current study investigated the relationship between fungal colonization, bacterial pathogens, inflammatory mediators, and salivary function in a pediatric group, aiming to clarify the connections between oral candidiasis and its periodontal effects. By combining clinical measurements with salivary biomarker analysis, our findings suggest that children with candidiasis exhibit a distinct biological profile, which may be useful for early diagnosis and risk assessment.

The observation that PI, GI, and BOP were significantly higher in the candidiasis group emphasizes how fungal colonization can worsen gingival inflammation. These findings support the idea that candidiasis in children is not just an isolated mucosal problem but part of a larger ecological shift within the oral cavity that impacts periodontal health [19]. Previous pediatric studies have primarily focused on dental caries [20] and malocclusion [21] as sources of inflammation; however, our results indicate that fungal overgrowth itself can lead to dysbiosis and tissue damage. The decreased salivary pH and buffer capacity in affected children further indicate that mucosal host defense mechanisms are compromised, creating an environment where pathogens can thrive.

The significant increase in *C. albicans* load, along with higher levels of *P. gingivalis*, *T. forsythia*, *T. denticola*, and *A. actinomycetemcomitans*, supports the idea that candidiasis and periodontal pathogens strengthen each other’s disease-causing potential [22]. In vitro studies have demonstrated that *Candida* can establish physical and metabolic interactions with bacterial biofilms, thereby enhancing bacterial survival and virulence [23,24]. The current findings in a pediatric group mirror these results, suggesting that co-colonization may be an early indicator of periodontal risk. This synergistic relationship may be particularly significant in children, whose immune system and oral microbiota are still developing, making them more susceptible to combined microbial challenges.

The cytokine profile observed—characterized by significant increases in IL-1β, IL-6, TNF-α, and IL-8—indicates that candidiasis in children is associated with an inflammatory state similar to that observed in early-onset periodontal disease. These cytokines are well-known for their roles in promoting the breakdown of connective tissue, stimulating osteoclast activity, and recruiting neutrophils [25]. Their strong correlation with both clinical indices and fungal load in our study suggests they could be reliable salivary biomarkers for active disease. The discovery of IL-1β as an independent predictor of candidiasis is significant, as this cytokine has been repeatedly linked to the progression from gingivitis to periodontitis.

The regression findings support the idea that pediatric gingival disease results from multiple, interacting factors. Logistic models revealed that *C. albicans* and IL-1β were significant predictors of candidiasis status, consistent with the role of fungal overgrowth and increased cytokine activity in exacerbating mucosal inflammation. Although some effect estimates were imprecise, the consistent associations across models strengthen the credibility of these findings.

In linear regression, the individual contributions of both *C. albicans* and IL-1β to gingival inflammation were clear, while salivary pH proved to be a strong protective factor. This finding aligns with previous research that a more acidic salivary environment promotes fungal persistence and microbial imbalance, ultimately increasing tissue vulnerability. The lesser role of flow rate may be due to its overlap with other physicochemical parameters, such as buffering capacity.

The PCA introduced a multivariate perspective, demonstrating that combining microbial and inflammatory markers can reliably distinguish between affected children and their healthy peers. The fact that *C. albicans* and cytokines heavily influenced PC1, while salivary defenses contributed in the opposite way, reflects the balance between pathogenic load and host protection that influences oral health. This multivariate approach emphasizes the potential of saliva-based panels, rather than individual biomarkers, to enhance diagnostic accuracy.

A key finding of our study is the consistent association between fungal colonization and gingival inflammation: *C. albicans* demonstrated strong correlations with the gingival index and was identified as an independent predictor in regression models. This observation aligns with recent studies suggesting that *Candida* should not be regarded solely as an opportunistic pathogen, but rather as an active participant in periodontal pathogenesis through fungal–bacterial interactions and local immune stimulation [26].

In addition, elevated levels of salivary IL-1β, IL-6, and TNF-α reflect activation of the local inflammatory response, which is consistent with previous studies on salivary biomarkers in periodontal disease. A recent study demonstrated that IL-1β is one of the most reliable salivary markers, strongly associated with disease severity and progression [27]. Another clinical study in adults confirmed significant associations between the severity of periodontal disease and salivary concentrations of IL-1β and TNF-α [28].

Salivary cytokines (IL-1β, IL-6, TNF-α, and IL-8) are elevated in various gingival inflammatory conditions and should be viewed as nonspecific markers of oral inflammation rather than definitive indicators of disease. In pediatric groups, increases in these mediators correlate with higher PI, GI, and BOP, suggesting a more intense biofilm–host interaction. Therefore, our results are interpreted using a panel approach: organism-specific qPCR (e.g., *Candida* ITS and particular periodontal targets) combined with salivary physicochemical properties (pH, unstimulated flow) provides better discrimination than any single cytokine. This approach aligns with growing evidence that multimarker profiles—combining microbial and host-response data—are more effective than isolated markers when differentiating overlapping mucosal inflammation phenotypes in children. Our previous pediatric trial in North-East Romania demonstrated that an adjunctive anti-inflammatory/antimicrobial topical treatment (Aloe vera) enhanced fungal clearance and adherence, indicating that microbiological and clinical outcomes often improve together under effective therapy. This supports the current multivariate salivary signature [29]. Additionally, a regional cross-sectional study found lower systemic nutritional biomarkers in children with candidiasis, suggesting that host nutritional status influences mucosal vulnerability and may shape the inflammatory environment reflected by salivary markers [30]. Overall, these findings highlight that salivary biomarkers are most useful when viewed as combined, contextual indicators that integrate organism burden, local host response, and salivary environment rather than as disease-specific markers.

Our observation that salivary pH and flow rate exert protective effects (inverse correlations and negative regression coefficients) adds a novel dimension. Reduced salivary pH and impaired clearance favor fungal persistence and bacterial overgrowth. This is supported by evidence showing that an acidic oral environment reflects an altered ecological balance in which pathogenic biofilms can thrive [31].

From a clinical perspective, these findings highlight the importance of early detection of children at risk for candidiasis-related periodontal inflammation. Integrative diagnostics that incorporate salivary biomarkers could improve routine pediatric dental care by allowing timely preventive actions. This is particularly vital in populations with limited access to dental services, poor nutrition, or compromised oral hygiene, where candidiasis and periodontal inflammation might go unnoticed. Adding simple salivary tests to school-based or community programs could serve as a cost-effective screening method.

The results of this investigation led to the rejection of the null hypothesis, as significant differences were found between children with oral candidiasis and healthy controls in both clinical and salivary biomarkers. The observed increases in *C. albicans* load, pro-inflammatory cytokines, and clinical inflammation indices (PI, GI, BOP), along with reduced salivary pH and flow, indicate a distinct biological profile linked to the disease. Moreover, the strong correlations and multivariable relationships between microbial burden and host inflammatory mediators highlight the interconnected nature of these factors in influencing oral health. These findings collectively disprove the idea that the parameters would not differ between groups or be mutually related. Instead, they support a model where fungal and bacterial colonization, increased cytokine activity, and weakened salivary defenses work together to promote mucosal inflammation and early periodontal issues in children.

Several limitations deserve attention. The cross-sectional design limits conclusions about causality, and it remains unclear whether fungal colonization comes before bacterial overgrowth or if both happen simultaneously. Although the sample size was enough to show apparent differences, long-term studies are needed to monitor biomarker changes over time and assess their ability to predict disease progression. Because buffering capacity may show higher test–retest reliability during stimulated flow in some protocols, our unstimulated measurements should be considered standardized comparative indices rather than definitive diagnostic cut-offs.

Additionally, while qPCR and cytokine assays provide strong quantification, combining these methods with metagenomic sequencing and proteomic profiling could offer a more comprehensive understanding of host–microbe interactions in pediatric oral disease.

Moreover, our protocol did not include measuring systemic immune or metabolic biomarkers, such as serum cytokines or HbA1c, nor did we quantify dietary intake or salivary glucose levels. As a result, unmeasured systemic and nutritional factors might have affected variability, and causal pathways cannot be established. Future longitudinal studies that incorporate systemic immune/metabolic profiling, salivary glucose measurements, and standardized dietary assessments are recommended.

Taken together, our findings reinforce the evidence that fungal–bacterial synergy, combined with an exaggerated host inflammatory response, contributes to the development of candidiasis-related gingival inflammation in children. Salivary profiling, which includes microbial, immunological, and physicochemical parameters, emerges as a promising non-invasive approach for diagnosis and monitoring. These insights open the door for targeted preventive strategies and promote a broader consideration of fungal pathogens within the conceptual framework of periodontal disease in pediatric populations.

## 5. Conclusions

This study reveals that pediatric oral candidiasis exhibits a distinct pattern of microbial colonization, inflammatory response, and salivary alterations. Higher levels of *Candida albicans* and periodontal bacteria in saliva, along with increased IL-1β, IL-6, TNF-α, and IL-8, are strongly associated with gingival inflammation and bleeding. Conversely, higher salivary pH and flow rate appear to have a protective effect. 

From a clinical perspective, these findings emphasize the potential of saliva as a non-invasive diagnostic tool for early detection and risk assessment in children. Incorporating salivary biomarker testing into routine pediatric dental care could enable more timely and personalized preventive strategies. Future long-term studies are necessary to validate these results and develop standardized biomarker panels for clinical application.

## Figures and Tables

**Figure 1 biomedicines-13-02837-f001:**
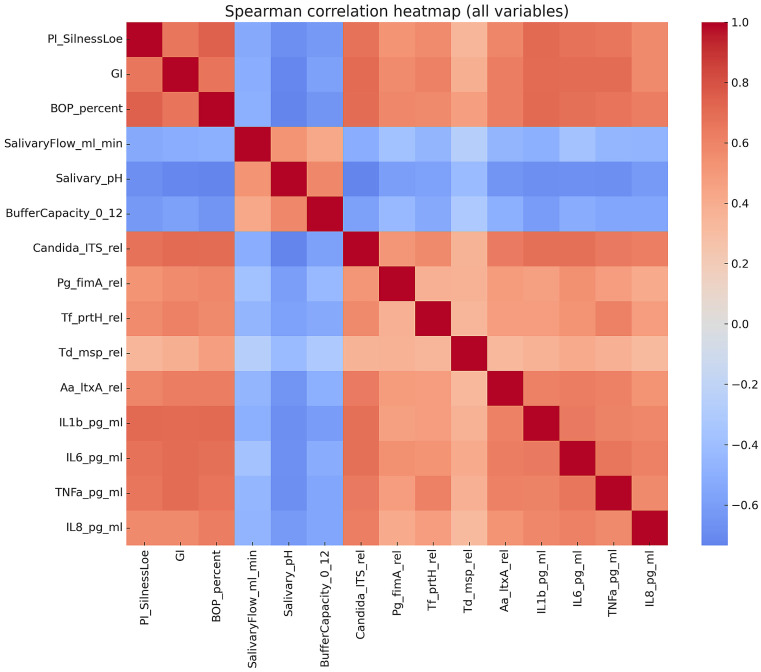
Heatmap of Spearman’s correlations between clinical indices, microbial and cytokine levels, and salivary parameters. Positive associations were observed between pathogens and cytokines and inflammation, while salivary pH and flow exhibited inverse correlations.

**Figure 2 biomedicines-13-02837-f002:**
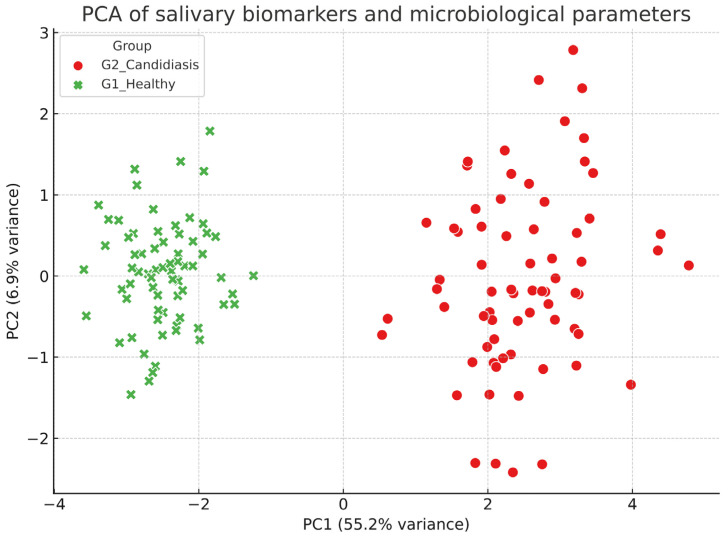
PCA demonstrated clear group separation (cumulative variance 62.1%), primarily along PC1, driven by *Candida albicans* and pro-inflammatory cytokines, with salivary pH and flow showing opposite contributions.

**Table 1 biomedicines-13-02837-t001:** Primer sequences used for qPCR detection of *Candida albicans* and major periodontal pathogens.

Target Organism	Gene (Marker)	Forward Primer (5′ → 3′)	Reverse Primer (5′ → 3′)
*Candida albicans*	ITS region	5′-TTG GTC CGT TCA TCG ATG AAG-3′	5′-GCT GCG TTC TTC ATC GAT GC-3′
*Porphyromonas gingivalis*	*fimA*	5′-TGG TAA AGG TCT TCG TGT TGT G-3′	5′-GGA GAA GGA GAA GAG TGG AGA-3′
*Tannerella forsythia*	*prtH*	5′-AGC TGT TCG GTT TAT GCG TTA G-3′	5′-TTC GGA TAA TCA GCA CCA TGC-3′
*Treponema denticola*	*msp*	5′-TGC TTA CAA GGA GGA GCT AAG-3′	5′-CTG TGT TCA GTT CAG GGA TTG-3′
*Aggregatibacter actinomycetemcomitans*	*ltxA*	5′-TGA TGC TGT TGA TTC GTT GAA-3′	5′-AAC CAA TGC CTC CAA ACA TG-3′
Human reference gene	*β-actin*	5′-CCT GGC ACC CAG CAC AAT-3′	5′-GCC GAT CCA CAC GGA GTA CT-3′

**Table 2 biomedicines-13-02837-t002:** Demographic and clinical characteristics of participants.

Parameter	Healthy (n = 70)	Candidiasis (n = 70)	*p*-Value
Age (years)	11.2 ± 2.1	11.5 ± 2.3	0.52
Male (%)	51.4%	48.6%	0.73
PI	0.72 ± 0.20	1.15 ± 0.25	<0.001
GI	0.68 ± 0.18	1.23 ± 0.22	<0.001
BOP (%)	12.4 ± 5.1	28.9 ± 7.8	<0.001
Salivary flow (mL/min)	0.89 ± 0.23	0.63 ± 0.19	<0.001
Salivary pH	7.23 ± 0.21	6.81 ± 0.24	<0.001
Buffer capacity	9.8 ± 1.1	7.1 ± 1.0	<0.001

**Table 3 biomedicines-13-02837-t003:** Salivary microbiological profiles.

Pathogen	Healthy (n = 70)	Candidiasis (n = 70)	*p*-Value
*Candida albicans* (ITS, rel. units)	1.2 ± 0.6	4.8 ± 1.7	<0.001
*Porphyromonas gingivalis* (*fimA*)	0.8 ± 0.4	2.1 ± 0.8	<0.001
*Tannerella forsythia* (*prtH*)	0.7 ± 0.3	1.9 ± 0.7	<0.001
*Treponema denticola* (*msp*)	0.6 ± 0.3	1.8 ± 0.6	<0.001
*Aggregatibacter actinomycetemcomitans* (*ltxA*)	0.5 ± 0.3	1.5 ± 0.5	<0.001

**Table 4 biomedicines-13-02837-t004:** Salivary cytokine concentrations.

Cytokine (pg/mL)	Healthy (n = 70)	Candidiasis (n = 70)	*p*-Value
IL-1β	15.2 ± 5.8	34.5 ± 9.1	<0.001
IL-6	12.4 ± 4.9	28.9 ± 8.3	<0.001
TNF-α	18.7 ± 6.2	36.8 ± 10.2	<0.001
IL-8	22.1 ± 7.3	41.5 ± 11.5	<0.001

**Table 5 biomedicines-13-02837-t005:** Logistic regression for candidiasis status (N = 140). Continuous predictors were standardized (OR per SD). Model fitted with median imputation for missing values, z-scaling, and L2-penalized logistic regression; 95% CIs by bootstrap (B = 2000). AUC = 1.00.

Predictor	Adjusted OR (Per SD)	95% CI	*p* (Two-Sided, Bootstrap)
*Candida* ITS (per SD)	3.58	2.37–5.67	<0.001
IL-1β (per SD)	6.75	4.44–9.53	<0.001
Salivary pH (per SD)	0.09	0.07–0.13	<0.001
Flow (per SD)	0.47	0.32–0.66	<0.001

**Table 6 biomedicines-13-02837-t006:** Linear regression for predictors of gingival index (dependent variable: GI).

Variable	Coef.	Std. Err.	t	*p* > |t|	[0.025	0.975]
const	5.726	1.285	4.458	0.0	3.186	8.267
*Candida*_ITS_rel	0.316	0.1	3.177	0.002	0.119	0.513
IL1b_pg_ml	0.013	0.003	3.642	0.0	0.006	0.02
Salivary_pH	−0.71	0.18	−3.94	0.0	−1.067	−0.354
SalivaryFlow_ml_min	−0.365	0.305	−1.198	0.233	−0.967	0.237

## Data Availability

The data used to support the findings of this study are available from the corresponding author upon request.

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
