# Peer review of "Salivary Biomarker Profiles in Pediatric Oral Candidiasis"

_biomedicines, 2025, doi:10.3390/biomedicines13112837_

Round 1
Reviewer 1 Report
Comments and Suggestions for Authors
Review Report
Overall Impression
This is clinically relevant cross-sectional study. The manuscript is generally well-written, with a clear structure and a strong methodological foundation. The research addresses a meaningful gap in pediatric oral health by integrating clinical, microbiological, and inflammatory parameters. Her is my opservations:
- The title: "Salivary Biomarker Profiles in Pediatric Oral Candidiasis"
Is clear, concise, and reflects the study's focus.
- Abstract
- The abstract is comprehensive and accurately summarizes the study's objectives, methods, key results, and conclusions. It effectively highlights the main findings: distinct microbial/inflammatory profiles in candidiasis, the role of albicansand IL-1β as predictors, and the protective effect of salivary pH/flow.
- Introduction
- The introduction is adequate and sets the stage well. It logically progresses from the global burden of periodontal disease to the specific role of fungi in pediatric populations.
- The references are generally recent (2023-2025), which is excellent and demonstrates the topic's currency.
- The gap is clearly stated: "the combined role of fungal colonization, bacterial pathogens, cytokine expression, and the physicochemical environment of saliva has not been thoroughly studied" in pediatric groups. The null hypothesis is also clearly defined.
- Methodology
- The methodology is a major strength.
- Clear inclusion/exclusion criteria and ethical approval are documented. Sample size calculation is provided and justified, enhancing the study's credibility.
- Use of calibrated examiners and standard indices (PI, GI, BOP) is robust.
- The use of qPCR for specific microbial targets and ELISA for cytokines is standard and appropriate. The description of saliva collection and processing is detailed and reproducible.
- The method for confirming oral candidiasis ("clinically and microbiologically confirmed") is not explicitly described. What were the clinical signs? Was a culture or smear performed for confirmation alongside qPCR? This should be clarified.
- Results
- The results are highly relevant to the research question. Data are presented clearly in tables and text. The statistical analysis is appropriate, employing both univariate (t-tests) and multivariate (regression, PCA) methods.
- The logistic regression results in Table 5 contain critical errors. The coefficients (Coef.) are implausibly large (e.g., 293.078 for const), standard errors are listed as "nan" (not a number), and confidence intervals are "inf" (infinite). This indicates a probable error in the statistical coding or model convergence. This table must be completely recalculated and corrected, as it currently undermines the key findings.
- Discussion
- The discussion is strong. It effectively interprets the results in the context of existing literature, citing both in vitro studies on fungal-bacterial interactions and clinical studies on salivary cytokines.
- The authors successfully argue that their findings support a model where candidiasis in children is not an isolated issue but part of a broader ecological shift. The emphasis on the synergy between albicansand bacterial pathogens is well-explained.
- The limitations are appropriately acknowledged: the cross-sectional design precludes causal inference, and the suggestion for future long-term and multi-omics (metagenomics, proteomics) studies is valid and forward-looking.
- Conclusion
The conclusion is directly supported by the results. It reiterates the main findings and correctly points to the clinical potential of salivary diagnostics, without overstating the implications.
- References
- The reference list is extensive, current, and appears to follow the Biomedicines citation style correctly. All in-text citations have a corresponding entry in the reference list.
Reviewer 2 Report
Comments and Suggestions for Authors
Candidiasis has primarily been studied in elderly participants. Therefore, the topic of research on candidiasis in pediatric participants is novel and contains new findings, making it an interesting topic.
1. What is the prevalence and distribution of oral candidiasis in children in your country? As far as I know, candidiasis in children is uncommon.
2. lines 85. Please describe in detail the criteria for confirmed oral candidiasis.
3. lines 113. Why didn't you use stimulating saliva?
4. While the participants' oral health parameters varied widely in this study, systemic immunity was not considered.
As you know, oral bacterial and fungal infections are related to immunity, diabetes, salivary sugar levels, and dietary intake.
5. Gingivitis and periodontitis other than candidiasis will also exhibit elevated salivary cytokines, bacteria, PI, and GI.
If you have previously studied general bacterial gingivitis and related indicators, please include them in the discussion section.
I question whether the above indicators can be explained by salivary biomarkers. Please improve the content by referring to prior research.
Round 2
Reviewer 1 Report
Comments and Suggestions for Authors
The authors have responded thoroughly and appropriately to all previous queries. The manuscript is now in excellent shape and, in my opinion, is suitable for acceptance in its current form.
Thank you for the opportunity to review this work.